# Allometric Equations for Predicting *Agave lechuguilla* Torr. Aboveground Biomass in Mexico

**Cristóbal de J. Flores-Hernández** [1] , **Jorge Méndez-González** [1,*], **Félix de J. Sánchez-Pérez** [2], **Fátima M. Méndez-Encina** [1], **Óscar M. López-Díaz** [1] **and Pablito M. López-Serrano** [3]

1   Forest Department, Autonomous Agrarian University Antonio Narro, Coahuila P. C. 25315, Mexico; cristobalf88@gmail.com (C.d.J.F.-H.); fatyencina12@gmail.com (F.M.M.-E.); menyld_22@hotmail.com (Ó.M.L.-D.)
2   Independent Statistical Consultant, Coahuila P. C. 25315, Mexico; fel1925@yahoo.com
3   Institute of Forestry and Wood Industry, Juarez University of the State of Durango, Durango P. C. 34120, Mexico; pmslopez@gmail.com
*   Correspondence: jmendezg@hotmail.com

**Abstract:** Quantifying biomass is important for determining the carbon stores in land ecosystems. The objective of this study was to predict aboveground biomass (AGB) of *Agave lechuguilla* Torr., in the states of Coahuila (*Coah*), San Luis Potosí (*SLP*) and Zacatecas (*Zac*), Mexico. To quantify AGB, we applied the direct method, selecting and harvesting representative plants from 32 sampling sites. To predict AGB, the potential and the Schumacher–Hall equations were tested using the ordinary least squares method using the average crown diameter (Cd) and total plant height (Ht) as predictors. Selection of the best model was based on coefficient of determination ($R^2$ adj.), standard error ($S_{xy}$), and the Akaike information criterion (AIC). Studentized residues, atypical observations, influential data, normality, variance homogeneity, and independence of errors were also analyzed. To validate the models, the statistic prediction error sum of squares (PRESS) was used. Moreover, dummy variables were included to define the existence of a global model. A total of 533 *A. lechuguilla* plants were sampled. The highest AGB was 8.17 kg; the plant heights varied from 3.50 cm to 118.00 cm. The Schumacher–Hall equation had the best statistics ($R^2$ adj. = 0.77, Sxy = 0.418, PRESS = 102.25, AIC = 632.2), but the dummy variables revealed different populations of this species, that is, an equation for each state. Satisfying the regression model assumptions assures that the predictions of *A. lechuguilla AGB* are robust and efficient, and thus able to quantify carbon reserves of the arid and semiarid regions of Mexico.

**Keywords:** allometric equations; aboveground biomass; Schumacher–Hall; dummy variables; robust regression

## 1. Introduction

Climate change is a problem of great magnitude. From 1950 to 2014, $CO_2$ emissions increased from 310 to 400 ppm [1], and currently the concentration is 409.92 ppm [2]. Forest ecosystems play an important role in regulating the climate by absorbing carbon [3] into plant biomass through photosynthesis [4]. Like many countries, Mexico has developed public policies to improve the well-being of the communities that live in arid regions. It has also developed a National Strategy for Reducing Emissions due to Deforestation and forest Degradation (ENAREDD+, the initials in Spanish), which aims decelerate, stop, and reverse loss of forest cover and increase carbon sequestering and ecosystem services through sustainable management [5], to contribute to climate change mitigation. Arid regions are the most vulnerable to climate change [6]. Conservation and improvement of these

areas is an option for maintaining carbon reserves [7]. Of Mexico's territory, 54% is arid, and more than 40% of the population live in these regions; however, it is known that above (23.2 Mg ha$^{-1}$ ± 4.15 Mg ha$^{-1}$) and underground biomass (11.2 Mg ha$^{-1}$ ± 3.54 Mg ha$^{-1}$) is more than the average (2 Mg ha$^{-1}$ to 5 Mg ha$^{-1}$) that exists in the world's deserts [8].

The UN Framework on Climate Change considers plant biomass as a necessary variable for prediction and mitigation of climate change [9]. Quantifying plant biomass allows us to determine how much carbon is stored in ecosystems [10]. For this reason, aboveground biomass equations are fundamental to evaluating carbon stores [11]. Aboveground biomass of some scrub vegetation and grasslands in arid and semiarid regions of Mexico varies from 1.6 to 30 Mg ha$^{-1}$, while stored carbon varies from 1 to 15.5 Mg C ha$^{-1}$ [12]. *Agave lechuguilla* Torr., locally called "lechuguilla", is distributed in arid and semiarid regions of the Chihuahua desert in Mexico and southern USA [13]. From this plant a fiber known as "ixtle" is extracted; ixtle is used to make brushes, mats, bags, and many other industrial products [14].

Aboveground biomass in plant species has been estimated by: (a) direct methods, in which complete plants are harvested to obtain fresh and dry weight [15]; and (b) by indirect methods, allometric equations developed from the first method [11,16]. Recently, the two procedures have been combined: aboveground biomass is estimated with remote sensors, the estimation is validated in the field with direct measurements and, using different algorithms [17], such as ordinary least squares, random forest or support vector regression [18–20], and the equations are generated. This type of equation uses dasometric variables that are easily measured, such as tree diameter, total height, and crown diameter, and are correlated with biomass [15]. Its predictive capacity depends on whether the assumptions of regression, such as normality, homogeneity of variance, and independence, are satisfied [21–23]. In this type of model it is possible to include indicator or dummy variables, which are variables that take values of 0, 1, or −1 to indicate absence or presence of some categoric effect [24] and that are useful to differentiate a set of data that belong to independent samples and, in this way, identify one or more models [25,26]. This method has been shown to be effective for resolving compatibility of biomass estimations at different scales [27].

Evidence reveals a scarcity of models of aboveground biomass (AGB) for arid regions' species, except for the study by Pando-Moreno et al. [28] which estimated usable fiber. Others have calculated the time of harvesting [29] or characterized the ecosystems where the plant grows [30,31]. The objective of our study was to fit two allometric equations using dummy variables for predicting aboveground biomass of *A. lechuguilla* samples from the Mexican states of Coahuila, Zacatecas, and San Luis Potosí.

## 2. Materials and Methods

### 2.1. Study Area

The study was conducted in north-central Mexico (99°48′–103° W, 22°50′–26°38′ N) in three states (Figure 1): Coahuila (*Coah*); San Luís Potosí (*SLP*); and Zacatecas (*Zac*). The predominant vegetation is rosetophile scrub and, in a lesser proportion, microphile scrub [32,33], made up mostly of *Agave lechuguilla* and genera such as *Dasylirion* sp., *Larrea* sp., *Opuntia* sp., *Acacia* sp., and species of the family Cactaceae [30,33]. Altitude varies from 810–2166 m; climate type is very arid and warm (BWhw) and arid temperate (BSokw). Mean annual precipitation varies from 126–600 mm and mean annual temperature is 16–22 °C [34].

### 2.2. Sampling of Aboveground Biomass

Based on Picard et al. [15], the direct method was applied to quantify the AGB of *A. lechuguilla*. Sampling was conducted only in "ejidos" and communities with permission to exploit plants of this species. For sample collection, the central part of the managed area was located and a radius of approximately 1 km was drawn; *n* number of plants were selected considering all the existing sizes (diameter, height and crow diameter). Each plant was measured for crown diameter (*Cd*) and total

height (*H*) with a 3 m tape measure (Truper®). The selected plants were then cut (only the aerial part) with manual tools to obtain fresh weight immediately on a 5 kg capacity Torrey® (model L-EQ) scale with a precision of 1 g. A subsample of all these aerial components was taken from each plant, with which its fresh weight was determined. The subsamples were taken to the laboratory and dried in an electric oven (mark Thermo Scientific™ HERAtherm™, model OMH750) at 70 °C until a constant weight was obtained. With this data, the ratio between dry and fresh weight was determined and multiplied by the total fresh weight per plant to obtain total dry biomass or *AGB* [11,15].

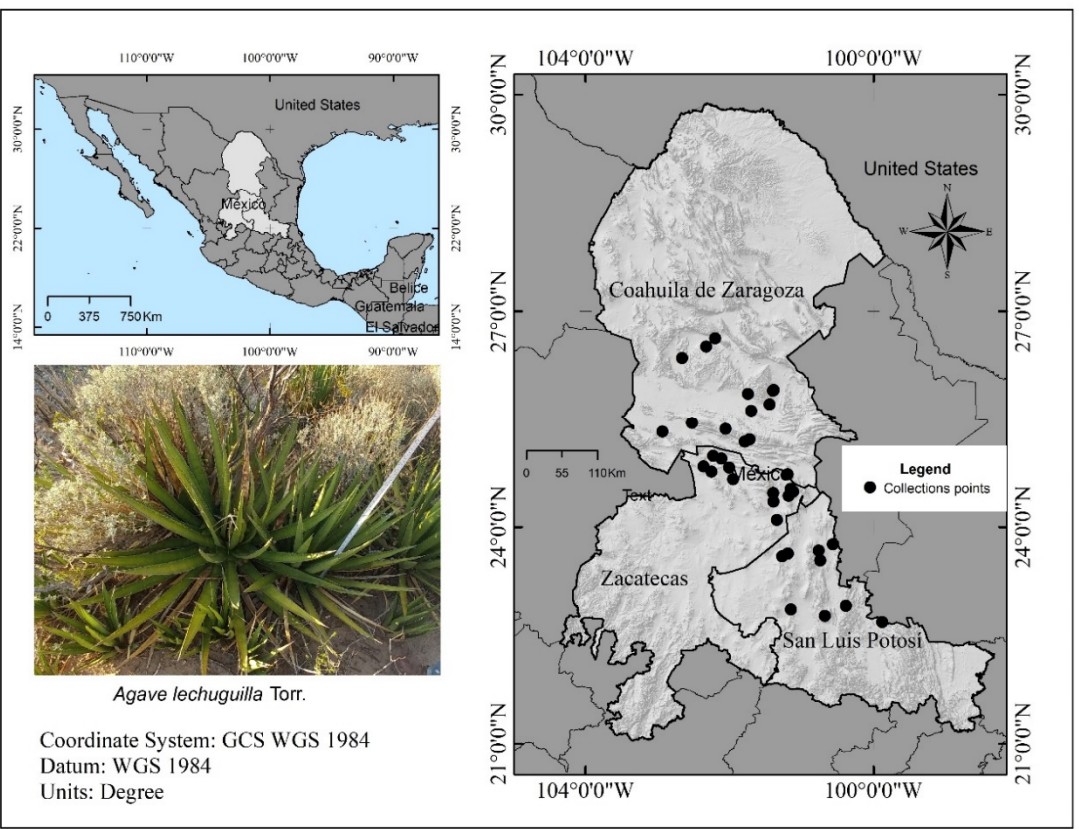

**Figure 1.** Location of collection points for *Agave lechuguilla* Torr. in Mexico.

## 2.3. Statistical Analysis

To determine *A. lechuguilla AGB*, the potential (Equation (1)) and the Schumacher–Hall [35] (Equation (2)) equations were used, both in their linearized form. These equations have been demonstrated to fit well and give reliable predictions of *AGB* of different forest species [4,11,36,37].

$$\ln(AGB_{Potential}) = \ln\beta_0 + \ln\beta_1 \times \ln(Cd) + \varepsilon \tag{1}$$

$$\ln(AGB_{Schumacher-Hall}) = \ln\beta_0 + \beta_1 \times \ln(Cd) + \beta_2 \times \ln(H) + \varepsilon \tag{2}$$

where *AGB* is dry aboveground biomass (kg), *Cd* is the average crown diameter (cm), *H* plant height (cm), ln natural logarithm, $\beta_{ij}$ are regression coefficients, and $\varepsilon$ the random error of the model.

To determine the existence of one or more models, dummy variables were added thus denoting the states, and leaving k−1 dummy variable based on the proposal of Montgomery et al. [22] and Fox [25]. An affectation to the intercept was considered (without interaction of the quantitative dummy variable), and the slope of the model (quantitative dummy interaction), resulting in Equations (3) and (4).

$$\begin{aligned}\ln(AGB_{Potential}) &= \ln\beta_0 + \beta_1\ln(Cd) + \beta_2\,(Zac) + \beta_3\,(SLP) + \beta_4\,(Zac:Cd) \\ &\quad + \beta_5\,(SLP:Cd) + \varepsilon\end{aligned} \tag{3}$$

$$
\begin{aligned}
\ln(AGB_{Schumacher-Hall}) \quad &= \ln \beta_0 + \beta_1 \, ln(Cd) + \beta_2 \, ln(H) + \beta_3 \, (Zac) + \beta_4 \, (SLP) \\
&+ \beta_5 \, (Zac:Cd) + \beta_6 \, (Zac:H) + \beta_7 \, (SLP:Cd) + \beta_8 \, (SLP:H) + \varepsilon
\end{aligned}
\tag{4}
$$

Statistical analyses were performed in R [38]. The models were fit using the ordinary least squares (OLS) method [22] using the 'stats' library and applying the correction factor by logarithmic transformation [39]. This method is widely used because the statistical–mathematical solution to obtain the regression coefficients is simple, minimizing the sum of squares of the observations with a linear relationship. Furthermore, this method allows the basic assumptions of a regression model to be met [25,26]; likewise, the use of logarithms by linearization of the model reduces the variance of the error and corrects model inadequacies [22,40]. Selection of the best model was based on the best coefficient of determination ($R^2$ adj.), the smallest standard error ($S_{xy}$), and value of the Akaike information criterion (AIC). To validate the models, the statistic prediction error sum of squares (PRESS), calculated with the 'qpcR' library [41], was used. This statistic is considered a form of crossed validation, as well as a way to evaluate the predictive capacity of a regression model [42].

### 2.4. Adaptation of the Regression Model

Atypical data in a regression model cause the noncompliance of the regression assumptions [25,43] and, thus, bias in the predictions. Based on the studentized residues of the model, atypical observations were identified, and values equal to or greater than 3 were eliminated from the analysis. Simultaneously, influencing data were evaluated with the 'stats' library [44]. With the 'nortest' library [45], normality of residuals was diagnosed using the Lilliefors test; variance homogeneity was verified using the Breusch-Pagan test of the 'lmtest' library [46], and independence of errors was checked with the Ljung-Box test [44]. When considering the multiple model, collinearity diagnostic was performed using the variance inflation factor (VIF) of the 'stats' library and the condition number or index (CN) based on eigenvalues [22].

### 2.5. Robust Regression Techniques

When the assumptions of regression are not satisfied by the effects of heavier, or atypical, observations, methods alternate to OLS are employed [21]. These methods consist of reducing the effect of these observations, as robust regression techniques or, in its case, generalized least squares [21,22,47]. Here, we applied the method of high breaking point estimation (MM estimation) proposed by Yohai [48] and derived from the maximum likelihood method (M estimation) proposed by Huber [49], using robust scale estimation (S estimation). The second method, least trimmed squares (LTS), minimizes the sum of squares of the smallest k residues and tolerates a large quantity of atypical values [21], both analyzed with the 'robustbase' library [50]. The method of least absolute deviation (LAD) minimizes the sum of absolute residues [51] in the 'L1pack' library [52] and generalized least squares (GLS) that estimate the coefficients of regression through iterative processes, based on the maximum likelihood method [53] and analyzed in the 'nlme' library [54]. In addition, the mean square error (MSE) and the $R^2$ adj. (adjusted coefficient of determination) were examined, and the regression assumptions of each method were verified.

## 3. Results and Discussion

### 3.1. Descriptive Statistics Within Its Algorithm

We sampled 533 *A. lechuguilla* plants distributed in the following manner: 175 in *Coah*, 178 in *SLP*, and 180 in *Zac* (Supplementary Materials). Plant size varied (Table 1). Plant height varied from 3.50 to 118.00 cm, Pando-Moreno et al. [28] reported heights between 25 and 97 cm for the same species in Coahuila and Tamaulipas. However, the analysis of variance (AOV) and the least significant difference (LSD) test showed that there were no statistically significant differences in *Cd* ($p > 0.05$) and *H* ($p > 0.05$) among states, but for AGB there were significant differences ($p < 0.05$); the highest average was found in *SLP* (0.89 kg plant$^{-1}$). According to Nobel and Quero [13], biomass is distributed as follows: 60% in

leaves, 10% in core, and 4% in roots. The highest AGB registered for the species was 8.17 kg in *SLP*, followed by *Zac* with 2.91 kg, and 2.03 kg for *Coah*. Conti et al. [55] reported that aboveground biomass in eight species that grown in semi-arid conditions oscillates from 0.1 to 25.2 kg.

**Table 1.** *Agave lechuguilla* Torr. descriptive statistics of the variables evaluated in Mexico.

| Parameter | Coahuila (*n* = 175) | | | San Luis Potosí (*n* = 178) | | | Zacatecas (*n* = 180) | | |
|---|---|---|---|---|---|---|---|---|---|
| | *Cd* | *H* | *AGB* | *Cd* | *H* | *AGB* | *Cd* | *H* | *AGB* |
| Minimum | 7.50 | 9.00 | 0.01 | 5.10 | 6.10 | 0.00 | 3.50 | 3.50 | 0.00 |
| Maximum | 128.50 | 95.00 | 2.03 | 166.50 | 118.00 | 8.17 | 127.50 | 87.00 | 2.91 |
| Mean | 48.66 | 44.02 | 0.49 | 54.82 | 45.60 | 0.89 | 49.40 | 41.75 | 0.45 |
| Standard deviation | 25.90 | 16.90 | 0.47 | 36.24 | 23.50 | 1.28 | 29.32 | 17.89 | 0.54 |
| C.V. | 53.22 | 38.40 | 96.16 | 66.10 | 51.53 | 144.03 | 59.35 | 42.86 | 120.24 |

Note: *Cd* = average crown diameter (cm), *H* = total plant height (cm), *AGB* = dry aboveground biomass (kg), *C.V.* = coefficient of variation (%).

### 3.2. Model Fit and Detection of Atypical Observations

Estimation of the regression coefficients by OLS showed that, for Equation (3), the coefficient of regression for *Cd* ($\beta_1$) and the indicator for *Zac* ($\beta_2$) were statistically significant ($p < 0.0001$) (Table 2), determining an independent model for predicting *AGB* in *Zac* (Equation (3a)).

**Table 2.** Model statistics for predicting aboveground biomass of *Agave lechuguilla* Torr. in México.

| Equation | Estimator | Value | Sxy (β) | Value *t* | Pr (>|t|) | $R^2$ adj. | Sxy |
|---|---|---|---|---|---|---|---|
| 3 | $\beta_0$ | −8.722 | 0.132 | −65.87 | 0.0001 | | |
| | $\beta_1$ (ln *Cd*) | 2.001 | 0.035 | 57.866 | 0.0001 | 0.865 | 0.558 |
| | $\beta_2$ [*Zac*] | −0.301 | 0.051 | −5.897 | 0.0001 | | |
| 4 | $\beta_0$ | −10.183 | 0.155 | −65.665 | 0.0001 | | |
| | $\beta_1$ (ln *Cd*) | 1.108 | 0.071 | 15.657 | 0.0001 | | |
| | $\beta_2$ (ln *H*) | 1.285 | 0.093 | 13.89 | 0.0001 | 0.901 | 0.478 |
| | $\beta_3$ [*Zac*] | −0.178 | 0.051 | −3.481 | 0.0001 | | |
| | $\beta_4$ [*SLP*] | 0.127 | 0.051 | 2.496 | 0.0100 | | |

Note: $S_{xy}$ (β) = standard error of regression coefficients; *Cd* = average crown diameter (cm), *H* = total plant height (cm), $R^2$ adj. = adjusted coefficient of determination; $S_{xy}$ = standard error of the model; ln = natural logarithm; $\beta_{ij}$ = coefficients of regression.

$$AGB_{Zac} = (\beta_0 + \beta_2 \, [Zac]) + \beta_1 \times \ln(Cd) \tag{3a}$$

$$AGB_{Coah \, y \, SLP} = \beta_0 + \beta_1 \times \ln(Cd) \tag{3b}$$

In Equation (4) the coefficients for *Cd* ($\beta_1$), *H* ($\beta_2$) and the indicators *Zac* ($\beta_3$) and *SLP* ($\beta_4$) were significant ($p < 0.0001$) (Table 2), indicating a model for predicting *AGB* in each state (Equations (4a) to (4c)).

$$AGB_{SLP} = (\beta_0 + \beta_4 \, [SLP]) + \beta_1 \ln(Cd) + \beta_2 \ln(H) \tag{4a}$$

$$AGB_{Coah} = \beta_0 + \beta_1 \ln(Cd) + \beta_2 \ln(H) \tag{4b}$$

$$AGB_{Zac} = (\beta_0 + \beta_3 \, [Zac]) + \beta_1 \ln(Cd) + \beta_2 \ln(H) \tag{4c}$$

where *AGB* is dry aboveground biomass (kg), *Cd* is the average crown diameter (cm), *H* is total height (cm), $\beta_{ij}$ are coefficients of regression, and ln is the natural logarithm.

The use of dummy variables in the *AGB* equation showed differences among states, affecting the independent term ($\beta_0$). Aquino et al. [56] applied this type of dummy variable to differentiate groups of species (*Cupania dentata* DC., *Alchornea latifolia* Sw. and *Inga punctata* Willd) when they estimated

*AGB* in southern Oaxaca, Mexico. Also, Cortés et al. [57] estimated *AGB* in six species of the genus Quercus in Guanajuato, Mexico.

Analysis of studentized residuals detected five atypical data for Equation (3) and nine for Equation (4) (Figure 2a,b); these were eliminated, and the equations were again fit with 527 and 524 data points respectively (Figure 2c,d), resulting in new coefficients of regression (Table 3). Faraway [21] and Fox [25] have demonstrated that this type of observation tends to bias the variance in regression coefficients and, because the model assumptions are not satisfied, the prediction is not robust and affects the direction of the regression slope [43]. For this reason, it is necessary to assure that atypical observations are not the product of mistakes in capturing the information or of errors of measuring tools [58].

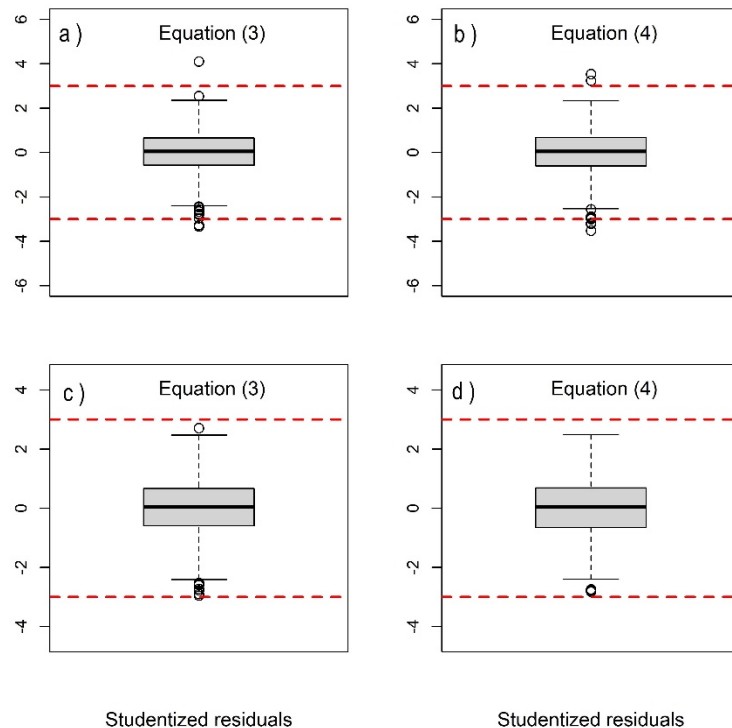

**Figure 2.** Atypical observations detected for the *Agave lechuguilla* Torr. aboveground biomass models in Mexico (**a**,**b**) and with no atypical observations for both equations (**c**,**d**).

**Table 3.** Statistics of the models (without outliers) for estimating aboveground biomass of *Agave lechuguilla* Torr. in Mexico.

| Equation | Estimator | Valor | IC | Pr (>\|t\|) | $R^2$ adj. | $S_{xy}$ | PRESS | AIC | CF |
|---|---|---|---|---|---|---|---|---|---|
| 3 | $\beta_0$ | −8.762 | (± 0.249) | 0.0001 | 0.877 | 0.531 | 149.55 | 834 | 1.151 |
| | $\beta_1$ (ln *Dp*) | 2.014 | (± 0.065) | 0.0001 | | | | | |
| | $\beta_2$ [*Zac*] | −0.299 | — | 0.0001 | | | | | |
| 4 | $\beta_0$ | −10.182 | (± 0.285) | 0.0001 | 0.914 | 0.44 | 102.25 | 632.2 | 1.101 |
| | $\beta_1$ (ln *Dp*) | 1.158 | (± 0.130) | 0.0001 | | | | | |
| | $\beta_2$ (ln *H*) | 1.236 | (± 0.169) | 0.0001 | | | | | |
| | $\beta_3$ [*Zac*] | −0.178 | — | 0.0001 | | | | | |
| | $\beta_4$ [*SLP*] | 0.143 | — | 0.01 | | | | | |

Note: $\beta_{ij}$ = coefficients of regression; *Cd* = average crown diameter (cm), *H* = total plant height (cm), CI = confidence interval (95 %); $R^2$ adj. = adjusted coefficient of determination; $S_{xy}$ = standard error; PRESS = prediction error sum of squares; AIC = akaike information criterion; CF = correction factor.

### 3.3. Selection of the Best Model

Equation (4) had the best statistics: significant coefficients of regression ($p < 0.0001$); smallest standard error ($S_{xy}$ = 0.439); and the lowest AIC (632), with an acceptable $R^2$ adj. (0.914). Moreover, it had a lower PRESS value (102.25), demonstrating the model's good predictive capacity [42], where *Cd* and *H* explain 91.4% of the variation in *A. lechuguilla AGB*. The correction factor (by transformation logarithmic) was 1.101 (Table 3). According to Ali et al. [59], the combination of these variables (*Cd* and *H*) and the use of the natural logarithm better predicts *AGB* in shrub species. For the above, this equation was selected to predict aboveground biomass by state (Figure 3). The Schumacher–Hall model showed good fit for estimating leaf biomass in arid region plants in Mexico, e.g., *Litsea parvifolia* (Hemsl.) and *Lippia graveolens* Kunth, with $R^2$ of 0.820 and 0.810, respectively [36,60], and total biomass in *Euphorbia antisyphilitica* Zucc, with $R^2$ of up to 0.897 [37] using *Cd*, base diameter and *H*. The $\beta_1$ coefficients reported in these studies (1.994, 1.935, and 1.611–1.703) were higher than that of *A. lechuguilla*, and $\beta_2$ coefficients (0.251, 0.257 y 0.300–0.774) were lower. Logically, these discrepancies are due to different *AGB* of each species. The usable fiber of *A. lechuguilla* can be estimated adequately with the polynomial model ($R^2$ = 0.869) using core height and diameter as predictor variables [28].

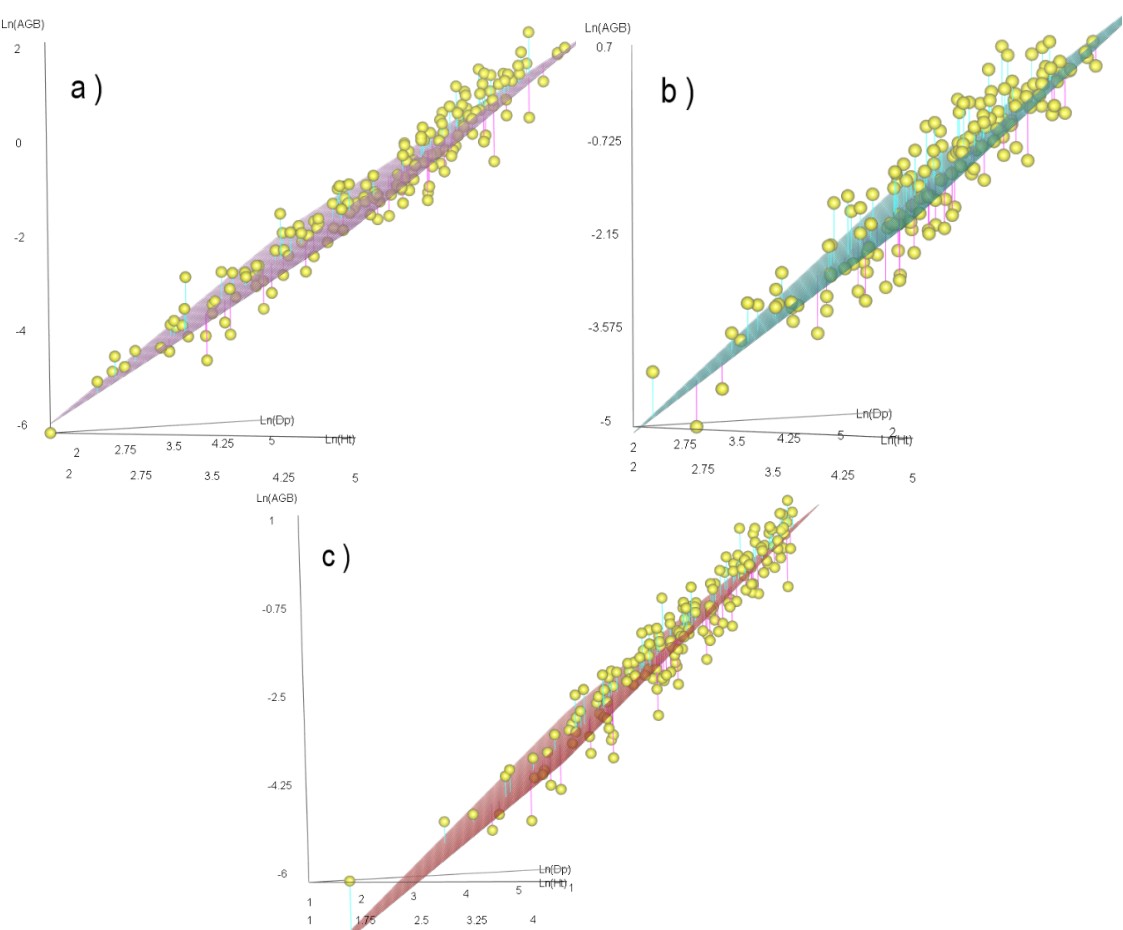

**Figure 3.** Tridimensional regression plane of Equations (4d) to (4f) for estimation of *Agave lechuguilla* Torr. aboveground biomass in Mexico. (**a**) San Luis Potosí, (**b**) Coahuila, (**c**) Zacatecas.

In tree species, Equation (4) has had good fit ($R^2$ adj. 0.971) for estimating aboveground biomass of *Quercus* sp. in Guanajuato, Mexico, using diameter at breast height (*Dbh*) and *H* as predictors [57], as with tropical species (*Cupania dentata* DC., *Alchornea latifolia* Sw. and *Inga punctata* Willd) with $R^2$ adj. > 0.98 [56]. Araujo et al. [61] used the same equation in 111 trees belonging to 50 tropical species in

restoration areas in Rio de Janeiro, Brazil, with an $R^2$ of only 0.65, a $\beta_1$ lower than that found in our study (0.91), and a higher $\beta_2$ (1.62), using diameter at breast height (*Dbh*) and height (*H*) as predictors.

Equation (3) explained only 87.70% of the *AGB* of *A. lechuguilla* and gave higher values of $S_{xy}$ (0.531) and AIC (834) than Equation (4). Equation (3) has been used to predict *AGB* in arid region species in Mexico, such as *Prosopis* sp. [11,62] and in central Asia in *Hardwickia binata* Roxb. [16] recording an $R^2$ between 0.70 and 0.99. In bushy species (*Calligonum polygonoides* L.) Singh and Singh [4] obtained an $R^2$ of 0.95; because of the nature of this plant, it was used as a predictor of biomass, the square root of the number of plants stems resulting in $\beta_1 = 3.065$, higher than that of our study ($\beta_1 = 2.014$). Ali et al. [59] used this same equation with eight shrub species and trees of less than 5 cm *Dbh* in a subtropical forest in China to estimate *AGB*, and obtained varying $R^2$ values (0.59–0.99). Conti et al. [55] developed equations for eight species of a semi-arid forest in Argentina, where they reported an $R^2$ of 0.61–0.85 for the multiple model using the variables crown, total height, number of branches, and diameter of the largest stem. In another study, Zeng et al. [63] used the quadratic model for total biomass prediction in four subtropical shrub species in China, obtaining an $R^2$ above 0.95 with the variables base diameter and height. This reveals the large diversity of allometric models used to quantify *AGB*, as well as the need to develop specific equations.

Three equations were derived from Equation (4) by effect of the dummy variables. By substituting the values of the coefficient of regression and adding the correction factor, we derived Equations (4d) to (4f) as follows:

$$AGB_{SLP} = \exp[-10.038 + 1.158 \times \ln(Cd) + 1.236 \times \ln(H)] \times 1.101 \tag{4d}$$

$$AGB_{Coah} = \exp[-10.182 + 1.158 \times \ln(Cd) + 1.236 \times \ln(H)] \times 1.101 \tag{4e}$$

$$AGB_{Zac} = \exp[-10.359 + 1.158 \times \ln(Cd) + 1.236 \times \ln(H)] \times 1.101 \tag{4f}$$

where *AGB* is dry aboveground biomass (kg), *Cd* is average crown diameter (cm), *H* is total height (cm), and ln is natural logarithm.

The relationship between observed *AGB* and estimated *AGB* had a strong linear correspondence (1:1 relation), with intercept ($\beta_0$) and slope ($\beta_1$) close to zero and one, respectively (*SLP*: $\beta_0 = -0.09$, $\beta_1 = 0.92$; *Coah*: $\beta_0 = -0.11$, $\beta_1 = 0.92$; *Zac*: $\beta_0 = -0.18$, $\beta_1 = 0.88$). This demonstrates the high predictive capacity of the model to estimate *AGB* of *A. lechuguilla* [23]. The technique has also been used with *E. antisyphilitica* [37], *Tamarindus indica* L. [64], and woody bush species of the Sonora Desert, Mexico [65]. According to Brown [66], one of the concerns in estimating biomass and carbon reserves is the possible error that can occur, from measuring different parameters to error in the final regression model. Cunia [67] attributes these possible errors to the sampling design. However, a review of the literature reveals that few comply with the assumptions of the regression models. In our study, we were very strict in this sense, and the estimations of *AGB* of *Agave lechuguilla* were efficient and robust.

### 3.4. Model Validation

The Lilliefors test in Equation (4) showed normality of residuals (D = 0.020, *p* value = 0.863) (Figure 4a). According to Fox [25], the coefficients of regression are efficient when this assumption is satisfied. Alonso and Montenegro [68] showed that satisfying normality improves estimations of the dependent variable. The Breusch–Pagan test denoted the residues as homoscedastic (BP = 4.580, df = 4, *p* = 0.333) (Figure 4b); when this assumption is not satisfied (heteroscedasticity), the coefficients are not efficient for any sample size [69]. The Ljung-Box test revealed that the residues are not correlated ($\chi^2 = 2.371$, df = 1, *p* = 0.124) (Figure 4c). These results suggest that the error variance is adequately estimated; the confidence intervals and the estimations of *A. lechuguilla AGB* are efficient and unbiased [22].

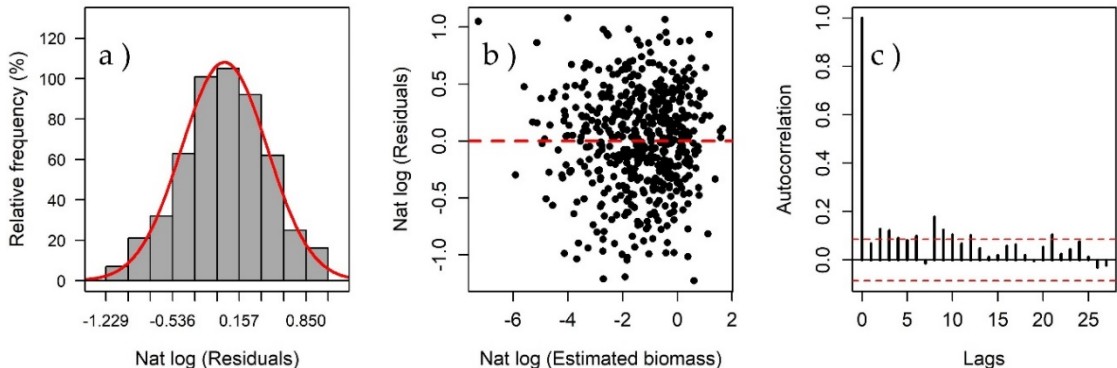

**Figure 4.** Histogram of residuals (**a**), estimated values against residuals (**b**) and correlogram of residuals of Equation (4) (**c**) of *Agave lechuguilla* Torr. in México.

In some cases, validation of equations is directed toward selection of the best statistics for their predictive capacity and, in the best of cases, to distribution of the residuals (e.g., Owate et al. [70]; Moore [71]; Zeng et al. [63]), leaving aside parametric tests that could verify regression assumptions and corroborate efficiency of the coefficients.

The H matrix, or Hat values, detected five potentially influencing observations, surpassing the critical value given by $h_{ii} = (3 \times k)/n = 0.029$, where k is the number of coefficients of regression and n the sample size (Figure 4a). The highest values are found in observations 354 and 357 ($h_{ii} = 0.05$ and 0.06, respectively). Therefore, this type of observation does not exert a leveraging effect on the regression slope [22]. The DFFITS (difference in fits) statistic identified three influencing observations ($p < 0.01$), surpassing the limit given by $2\sqrt{(k/n)} = 0.195$ (Figure 4b). These observations have a slight influence in *A. lechuguilla* AGB estimation [22,72]. Verification of influencing observations after fitting any equation provides relevant information when considering omission or not of an observation [26]. These observations were not removed from the database. Observation 354 is influential both potentially and in estimation, but it is distributed on the same cartesian plane as the rest of the observations (Figure 5).

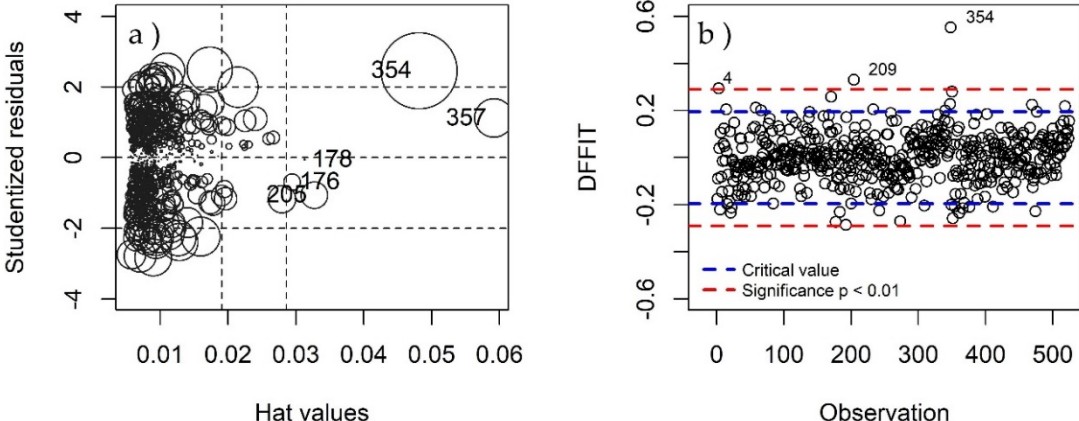

**Figure 5.** Detection of potentially influencing observations (**a**) and observations influential in estimation (**b**) of Equation (4) for estimation of aboveground biomass of *Agave lechuguilla* Torr. in México.

Equation (4) did not have serious collinearity; the VIF values of both variables were less than 10 (*Cd* = 5.64 and *H* = 5.66). VIF values above 10 in a regressor indicates problems of collinearity [47]; others consider a problem to exist when values are above 5 [22]. The condition number (CN) proposed by Montgomery et al. [22] demonstrated that there is moderate collinearity for *H* (CN = 193.85) and null collinearity for *Cd* (CN = 7.90). Therefore, application of a corrective method, such as ridge regression, principal components, or partial least squares, was not necessary [21].

### 3.5. Robust Estimation

Application of the robust techniques tested in Equation (4) to determine whether to conserve atypical data or not was highly significant ($p < 0.0001$) in the coefficient of regression (Table). The MM estimation technique reduced the confidence intervals (CI) of the regression coefficients ($\beta_0 = \pm 0.300$, $\beta_1 = \pm 0.136$ and $\beta_2 = \pm 0.177$), compared with the LAD, LTS, and GLS methods (Table 4). Moreover, this technique was the only one to satisfy the assumption of error normality, according to the Lilliferos test ($D = 0.039$, $p = 0.053$), and it had the smallest mean square error (MSE). Simpson and Montgomery [73] demonstrated that the MM estimation method is efficient when there are atypical data, but sensitive to high leverage atypical data [74].

**Table 4.** Statistics of robust regression methods.

| Estimator | OLS | MM | LAD | LTS | GLS |
|---|---|---|---|---|---|
| $\beta_0$ | −10.183 * (±0.304) | −10.214 * (±0.300) | −10.244 * (±0.467) | −10.349 * | −9.959 * (±0.328) |
| $\beta_1$ | 1.108 * (±0.139) | 1.129 * (±0.136) | 1.155 * (±0.214) | 1.181 * | 1.048 * (±0.138) |
| $\beta_2$ | 1.285 * (±0.181) | 1.273 * (±0.177) | 1.245 * (±0.279) | 1.208 * | 1.285 * (±0.180) |
| $\beta_3$ (*Zac*) | −0.178 * (±0.100) | −0.161 * (±0.098) | −0.099 ** (±0.072) | −0.098 * | −0.179 * (±0.113) |
| $\beta_4$ (*SLP*) | 0.127 ** (±0.100) | 0.147 * (±0.098) | 0.210 (±0.154) | 0.195 *** | 0.127 ** (±0.113) |
| $R^2$ adj. | 0.901 | 0.907 | 0.901 | 0.919 | 0.901 |
| MSE | 0.226 | 0.226 | 0.228 | 0.229 | 0.228 |
| CF | 1.121 | 1.111 | —— | 1.092 | 1.120 |
| Normality (LF) | 0.043 | 0.053 | 0.022 | 0.014 | 0.014 |
| Homogeneity (B-P) | 0.031 | 0.031 | 0.031 | 0.031 | —— |
| Autocorrelation (LJ-B) | 0.007 | 0.013 | 0.013 | 0.015 | 0.000 |

Note: OLS = Ordinary least squares; MM = MM estimation; LAD = Absolute least deviation; LTS = Least trimmed squares; GLS = Generalized least squares; MSE = Mean square error; CF = Correction factor; LF = Lilliefors test; B-P = Breusch-Pagan test; LJ-B = Ljoung-Box test; * = $p < 0.0001$; ** = $p < 0.001$; *** = $p < 0.01$.

When the above was demonstrated the MM estimation technique had a smaller MSE (0.226), while the LAD, LTS, and GLS methods had higher values (0.228, 0.229 and 0.228); according to Simpson and Montgomery [73], this parameter shows the efficiency of the robust method. Some studies have demonstrated the efficiency of robust techniques [75–77], specifically LTS, which predicts adequately when there are atypical observations [78]. That, however, depends on the nature of the observations [21]. Susanti et al. [79] demonstrated the efficiency of the S estimation technique against the M and MM techniques, significantly reducing the effect of atypical observations and increasing the value or $R^2$.

The $R^2$ was similar among the robust techniques (0.901 to 0.919). The highest value was found with the LTS technique (Table 4). According to Faraway [21], there is no sense in evaluating this coefficient in robust techniques; we calculated it only to make comparisons. However, Alma [74] compared four robust methods, of which the MM method stood out over the estimation methods M, LTS, and S estimation using $R^2$. The usefulness of robust estimation lies in the fact that atypical observations do not influence estimations of the coefficients of regression [73]. Also, they can be used to stabilize the variance [22] in the GLS method.

The estimators obtained using the MM technique are an alternative for predicting *A. lechuguilla* AGB by state considered the outliers detected. Derived from this, the following equations are proposed:

$$AGB_{SLP} = \exp[-10.067 + 1.129 \times \ln(Cd) + 1.273 \times \ln(H)] \times 1.11 \tag{5a}$$

$$AGB_{Coah} = \exp[-10.214 + 1.129 \times \ln(Cd) + 1.273 \times \ln(H)] \times 1.11 \tag{5b}$$

$$AGB_{Zac} = \exp[-10.375 + 1.129 \times \ln(Cd) + 1.273 \times \ln(H)] \times 1.11 \tag{5c}$$

## 4. Conclusions

The use of dummy variables in the Schumacher–Hall equation was found to be a useful tool to differentiate models of *A. lechuguilla* aboveground biomass at the regional scale and, therefore, to improve estimations of this species, resulting in a model of *AGB* by state. Satisfaction of the regression

model assumptions confirms that the Schumacher–Hall equation predictions of *A. lechuguilla AGB* are efficient and robust. When the method of ordinary least squares has difficulties in satisfying the assumptions because of atypical data, the robust regression technique is a good option, but it is necessary to test the different methods, since each method will give different results with different sets of data. The inclusion of the variables average crown diameter (*Cd*) and plant height (*H*) in the Schumacher–Hall model improved the predictions more than when a single variable was used (potential model). Crown diameter and plant height can predict up to 90% of the *AGB* of this species. The robust regression technique by MM estimation to predict the *AGB* of this species was shown to be a good alternative when atypical data are present. This study contributes an equation to predict *A. lechuguilla AGB* at a regional scale and, thus, to quantify carbon reserves of the arid and semiarid regions of Mexico.

**Supplementary Materials:** The following are available online at http://www.mdpi.com/1999-4907/11/7/784/s1.

**Author Contributions:** C.d.J.F.-H. performed statistical analysis and writing of the manuscript. J.M.-G. and F.d.J.S.-P. assisted in statistical analysis, writing and review. F.M.M.-E., Ó.M.L.-D. and P.M.L.-S. assisted in collecting information in the field. All authors have read and agreed to the published version of the manuscript.

**Funding:** The current study was funded by the Forestry National Commission (CONAFOR) and CONACYT, through project number: 2017-4-29267, titled "Best management practices and generation of volume and biomass equations for the main non-timber forest species of economic importance in the arid and semi-arid ecosystems of Mexico".

**Acknowledgments:** To the National Council of Science and Technology (CONACYT) for the scholarship awarded to the first author for postgraduate studies.

**Conflicts of Interest:** The authors declare that they have no conflict of interest.

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
