# Peer review of "Allometric Equations for Predicting Agave lechuguilla Torr. Aboveground Biomass in Mexico"

_forests, doi:10.3390/f11070784_

Round 1

Reviewer 1 Report

ALLOMETRIC EQUATIONS FOR PREDICTING Agave lechuguilla Torr. ABOVEGROUND BIOMASS IN MEXICO

Abstract

Line 27: Is the sentence well redacted?

  1. Introduction

Lines 53-60: The review is very poor. Nothing is explained about the different methodologies (MLR, RF, SVR), different combination of variables used by other authors.

Line 55: The sentence is correctly written?

Line 62-66: These lines should not be in the last paragraph of the introduction, a paragraph describing the species of the Mexican desert can be suitable, but earlier than the last paragraph, where the main goal and structure of the article should be explained.

  1. Material and Methods

Methods are, overall, very poorly described. A flowchart explaining the main processes would be suitable.

  • Study area

No characterization of the vegetation of the study area is done.

Line 77: Figure 1. The map with the biggest scale is not necessary. Mexico is not an isle, so the lands adjacent to it should appear is the map, gulfs and oceans included. The same thing happens with the map of the study area on the right side of the figure. Scale is missing in all the maps.

A map showing the distribution of the studied species in the study area would be suitable.

  • Sampling of aboveground biomass

There is no information about the shape and size of the plots, the number of plots included in the study should be mentioned in this point.

  • Statistical analysis

Why have you used this specific equations (1,2)? Explain the reasons.

More information about the dummy variables would be appropriate.

Why have you used the OLS method? Explain the reasons.

In this point an explanation about the transformation of the logarithmic values with a reference.

  • Adaptation of the regression model

Line 110: You should say that you are going to verify the existence of atypical observations, otherwise, it is a result. In case they exist, explain which tests or methodologies are you going to use.

  • Robust regression techniques

More information about this techniques should be given, a description of each methodology with references.

Line 121: What is MM? You have not mentioned it yet, so the first time write, please, the complete name and the the acronym.

Line 124: Define what MSE is and the same with R2.

A flowchart of the entire process would be appropriate.

  1. Results and discussion

  • Descriptive statistics

How many plots were in each area?

Line 138: The range of the biomass values in eight species in semi-arid conditions is not very wide?

3.4 Model validation

Cross-validation of the model has been carried out?

Sensitivity of the model has been analysed?

Exist other biomass values in the same are for the same species to compare with?

Inference of the model has been done?

A map showing the obtained biomass should be appropriate.

Line 296: What is M technique, you have not introduce it.

Line 301: Why does MM method stand out? Give reasons.

  1. Conclusions

The conclusions are too obvious, they must be improved.

Author Response

“ALLOMETRIC EQUATIONS FOR PREDICTING Agave lechuguilla Torr. ABOVEGROUND BIOMASS IN MEXICO”

Reviewer 1

Abstract

Line 27: Is the sentence well redacted?
The sentence was verified, it is correct

  1. Introduction

Lines 53-60: The review is very poor. Nothing is explained about the different methodologies (MLR, RF, SVR), different combination of variables used by other authors.

The review was expanded. Allusion is made to the suggested methods and new citations are attached (Lines 58-61)

Line 55: The sentence is correctly written?

The sentence is correct. The translator was consulted to verify the sentence.

Line 62-66: These lines should not be in the last paragraph of the introduction, a paragraph describing the species of the Mexican desert can be suitable, but earlier than the last paragraph, where the main goal and structure of the article should be explained.

The paragraph was moved. the manuscript shows the location (Lines 52-55)

  1. Material and Methods

Methods are, overall, very poorly described. A flowchart explaining the main processes would be suitable.

The authors consider that a flowchart did not provide information to the document, but a new information of methods are added.

  • Study area

No characterization of the vegetation of the study area is done.
A brief characterization of the existing vegetation in the study area is added. New authors are included (Lines 82-84)

Line 77: Figure 1. The map with the biggest scale is not necessary. Mexico is not an isle, so the lands adjacent to it should appear is the map, gulfs and oceans included. The same thing happens with the map of the study area on the right side of the figure. Scale is missing in all the maps.

The world map was removed, neighboring countries are annexed. As suggested, the maps are improved and the scale is included.

A map showing the distribution of the studied species in the study area would be suitable.

A distribution map of the species was searched, but was not found in the literature. The authors have developed maps of current species distribution (niche models, but only includes Mexico).

  • Sampling of aboveground biomass

There is no information about the shape and size of the plots, the number of plots included in the study should be mentioned in this point.

The work was carried out according to the proposal of the majority of the authors cited in the manuscript, for example: Brown et al., (1996), Návar et al., (2019), Picard et al (2012), Segura et al (2008), Cunia, (1987) etc. Sampling is selective, so no sampling plots are established, unless it was required to quantify biomass on a defined surface (usually small).

It is required to select individuals of all sizes or dimensions, so that the predictions of the model include all these categories and not those that have been registered in delimited plots. However, the observations of the referees were good, missing information was added. It is pointed out in the manuscript

  • Statistical analysis

Why have you used this specific equations (1,2)? Explain the reasons.

The justification for using this type of model is indicated in the manuscript (Lines 104-105)

More information about the dummy variables would be appropriate.

As requested, brief briefing in introduction was added to broaden this topic, new authors were added (Lines 65-69)

Why have you used the OLS method? Explain the reasons.

Information that justifies the use of the OLS method is provided (Lines 123-126)

In this point an explanation about the transformation of the logarithmic values with a reference.

Information that justifies the use of logarithmic is provided (Lines 126-127)

  • Adaptation of the regression model

Line 110: You should say that you are going to verify the existence of atypical observations, otherwise, it is a result. In case they exist, explain which tests or methodologies are you going to use.

This section adds more information in which it is explained that the non-compliance with the regression assumptions tends to bias the predictions, and therefore, the imperative need to analyze them and solve them. detailed information is provided (Lines 133-136).

  • Robust regression techniques

More information about this techniques should be given, a description of each methodology with references.

As suggested by the reviewers, information of each robust estimation method is added.

Line 121: What is MM? You have not mentioned it yet, so the first time write, please, the complete name and the the acronym.

With the information provided above, this suggestion is addressed (Lines 146-153)

Line 124: Define what MSE is and the same with R2.

Are indicated in the text. The last, was missing to define it (Lines 153-154)

A flowchart of the entire process would be appropriate.

A flow chart was made, however the authors will conjecture that it would duplicate the information provided in the text. It was not included.

  1. Results and discussion

  • Descriptive statistics

How many plots were in each area?

With the new information added in the methodology, this observation is addressed (Lines 90-94)

Line 138: The range of the biomass values in eight species in semi-arid conditions is not very wide?

The information is correct. These are different species of arid zones, of different dimensions, therefore contrasting differences in biomass values.

3.4 Model validation

Cross-validation of the model has been carried out?

At the suggestion of the reviewer, a new analysis was made in which the PRESS statistic was included, which is a way of making cross validation of a regression model (Lines 129-131)

Sensitivity of the model has been analysed?

It was not the objective of the investigation. Sensitivity analysis can be as extensive, from coefficients, regression methods, to statistical adjustment.

Exist other biomass values in the same are for the same species to compare with?
In review of the literature carried out by the authors, there are no papers on biomass models in this species to make comparisons.

Inference of the model has been done?

Inferences about the model included those indicated in the document. On the hypothesis tests of the regression coefficients, hypothesis tests on the independent variables and their statistical significance and the AOV (analysis of variance) of each model. In the same way the hypothesis tests of the analysis of outlier and outlier data

A map showing the obtained biomass should be appropriate.

We consider that a map is not convenient to represent the biomass estimates generated by the model. The objective was to generate a model to predict the aerial biomass of this species. Representing biomass on a map requires other types of methodologies, which were not considered here because it was not the object of study.

Line 296: What is M technique, you have not introduce it.

With the added observations, this concept is already explained (Lines146-148)

Line 301: Why does MM method stand out? Give reasons.

It is Explained in the manuscript (Lines 314-318)

  1. Conclusions

The conclusions are too obvious, they must be improved.

The conclusions were improved. They are indicated in the document

Reviewer 2 Report

The present study focuses on the parameterization of alometric equations for the calculation of above-ground biomass Agave lechuguilla Torr. A total of 533 Agave plants were included in the analysis. Samples were taken in the Coahuila, San Luis Potosí and Zacatecas areas. For the model parameterization model was use linear regression with dummy variables. Data were tranformed by natural logarithm. After the removal of the outliers, a new model was parameterized, including a correction factor for the elimination of the bias resulting from the logarithmic transformation. Next, the model validation was carried out.

I have the following recommendations for the study.

Figure 1. I recommend the legend in the image on the right to move to the bottom edge. The map shows that in SLP samples were taken on 9 plots, in ZAC on about 13 plots and in COAH on about 12 plots. However, there is no information about these plots in the text. For example coordinates – the number of plants taken per area, altitude, and plants parameters. I recommend fill this information in the form of a table.

Rows 79 – 80: In text is „selected representative Agave plants in each state“. There is no information on how the sample plants were selected (e.g. by average height, etc.).

Rows 83 – 84: From what part of the plant was the sample taken? In row 134 is: „we included all the components“. If you have included roots then it is not aboveground biomass but whole biomass. It can be assumed that the humidity of leaves, cores an roots are different. However, the methodology does not show how and from which parts of the plants the samples were taken or separated.

Rows 89 – 90: The input type of model is one dimension and multidimensional linear regression with logarithmic data transformation in my opinion. If you are using the „Schumacher Hall equation“, it would be a good idea to quote the source in the article.

Row 136 – 137: For comparing multiple data sets, t test is inappropriate due to high increase in the probability of a second type error. Variance analysis should be used to compare multiple groups and then post hoc test.

Row 137 – 138: comparing the weight of biomass with 9 species is very general and has no telling value

Table 1. I assume the weight is in the dry matter, but it is necessary to add it to the explanatory notes.

Figure 3. Axis labels are almost illegible, I recommend enlarging and adjusting the location.

The parameterization of the biomass calculation model is carried out at a very high level. After incorporation of comments, I recommend publishing the article in Forests.

Author Response

“ALLOMETRIC EQUATIONS FOR PREDICTING Agave lechuguilla Torr. ABOVEGROUND BIOMASS IN MEXICO”

Reviewer 2 

Figure 1. I recommend the legend in the image on the right to move to the bottom edge.

Improvements were made to the map. A photograph of the species was included.

The map shows that in SLP samples were taken on 9 plots, in ZAC on about 13 plots and in COAH on about 12 plots. In the methodology section this observation is justified. However, there is no information about these plots in the text. For example coordinates – the number of plants taken per area, altitude, and plants parameters. I recommend fill this information in the form of a table.

The general parameters of the plants are presented in table 1. The recommendation is met, an appendix is ​​added, important characteristics are included.

Rows 79 – 80: In text is „selected representative Agave plants in each state“. There is no information on how the sample plants were selected (e.g. by average height, etc.).

This observation was complemented by revision 1. It is explained in section 2.2. Sampling of aboveground biomass (Lines 90-94)

Rows 83 – 84: From what part of the plant was the sample taken? In row 134 is: „we included all the components“. If you have included roots then it is not aboveground biomass but whole biomass. It can be assumed that the humidity of leaves, cores an roots are different. However, the methodology does not show how and from which parts of the plants the samples were taken or separated.

Text where inclusion of all components is mentioned was removed due to confusion. In secction 2.2.            Sampling of aboveground biomass, it is explained that only the aerial part was sampled

Rows 89 – 90: The input type of model is one dimension and multidimensional linear regression with logarithmic data transformation in my opinion. If you are using the „Schumacher Hall equation“, it would be a good idea to quote the source in the article.

The osbervacion was attended. The corresponding citation of the equation was added.

Row 136 – 137: For comparing multiple data sets, t test is inappropriate due to high increase in the probability of a second type error. Variance analysis should be used to compare multiple groups and then post hoc test.

By decision of the co-authors, this analysis was removed from the manuscript.

Row 137 – 138: comparing the weight of biomass with 9 species is very general and has no telling value.

The reviewer is right, however due to the absence of biomass models in this species, this information was kept as a way to distinguish the amounts of biomsa between arid zonsa species.

Table 1. I assume the weight is in the dry matter, but it is necessary to add it to the explanatory notes.

A note to explain that it is dry matter was added.

Figure 3. Axis labels are almost illegible, I recommend enlarging and adjusting the location.

The code used to generate the graphics did not allow the size of the axes to be modified, however, the size of the graphics was expanded.
